# Evaluation of Current Studies to Elucidate Processes in Dental Follicle Cells Driving Osteogenic Differentiation

**DOI:** 10.3390/biomedicines11102787

**Published:** 2023-10-13

**Authors:** Christian Morsczeck, Michela De Pellegrin, Anja Reck, Torsten E. Reichert

**Affiliations:** Department of Oral and Maxillofacial Surgery, University Hospital Regensburg, Franz-Josef-Strauss-Allee 11, 93053 Regensburg, Germanyanja.reck@ukr.de (A.R.); torsten.reichert@ukr.de (T.E.R.)

**Keywords:** dental stem cells, lipidomics, metabolism, protein kinase C

## Abstract

When research on osteogenic differentiation in dental follicle cells (DFCs) began, projects focused on bone morphogenetic protein (BMP) signaling. The BMP pathway induces the transcription factor DLX3, whichh in turn induces the BMP signaling pathway via a positive feedback mechanism. However, this BMP2/DLX3 signaling pathway only seems to support the early phase of osteogenic differentiation, since simultaneous induction of BMP2 or DLX3 does not further promote differentiation. Recent data showed that inhibition of classical protein kinase C (PKCs) supports the mineralization of DFCs and that osteogenic differentiation is sensitive to changes in signaling pathways, such as protein kinase B (PKB), also known as AKT. Small changes in the lipidome seem to confirm the participation of AKT and PKC in osteogenic differentiation. In addition, metabolic processes, such as fatty acid biosynthesis, oxidative phosphorylation, or glycolysis, are essential for the osteogenic differentiation of DFCs. This review article attempts not only to bring the various factors into a coherent picture of osteogenic differentiation in DFCs, but also to relate them to recent developments in other types of osteogenic progenitor cells.

## 1. Introduction

Regenerative dentistry is an emerging area of dental research, and in recent years, numerous articles have appeared dealing with tooth development, tissue regeneration, organoids, and stem cell differentiation [1,2,3,4,5,6,7,8]. A tooth consists of three mineralized tissues derived from three different tooth germ tissues: enamel organ, dental papilla, and dental follicle. The dental follicle derives from the dental mesoderm, which originates from neural crest cells in early stages of mammalian development [9]. This tissue is not only involved in the development of the periodontium, but is also responsible for tooth eruption [10,11,12]. The dental follicle is collagenous tissue that is interspersed with small blood vessels. In the vicinity of these blood vessels, small cells express typical markers of dental stem cells, such as nestin [13,14,15]. For periodontal development, it contains progenitor cells for the mineralizing cells, dental cementoblasts, and alveolar osteoblasts, as well as for periodontal soft-tissue fibroblasts that connect the mineralized alveolar bone and dental cementum [16,17,18].

In contrast to the enamel organ, but similar to the dental papilla/pulp, the dental follicle can be obtained from wisdom tooth extractions, and so they were available for the isolation of somatic stem cells, which can be used for the regenerative dentistry [19,20]. In order to be able to isolate undifferentiated single cells, the tooth follicle separated from the extracted wisdom tooth not only has to be cut in small pieces, but also undergo further enzymatic treatment in order to dissolve the stable collagen matrix [19]. Isolated single dental follicle cells (DFCs) built small colonies consisting of at least 50 cells after 14 days of cell culture in a standard medium for fibroblasts, which is typical for stem cells and osteogenic progenitor cells and has been demonstrated in numerous previous approaches [21,22]. Typical markers of dental follicle cells are the intermediate filament protein nestin and the protein marker of mesenchymal stem cells, STRO-1 [14,19,23]. Differentiation approaches were performed to prove that these dental follicle cells are undifferentiated stem or precursor cells. For example, dental follicle cells could be differentiated into neural cells, which could be demonstrated by typical markers for neuronal cells [24]. Most importantly, however, was the demonstration that DFCs can be differentiated into cells of the periodontium, such as cementoblasts or alveolar osteoblasts, and for regenerative medicine [25,26,27,28]. Therefore, an osteogenic differentiation approach under in vitro conditions was started. Surprisingly, this experiment showed that dental follicle cells not only differentiate into cells that produce mineralized tissue, but also into fibroblast-like cells that produce connective tissue [13,19]. It was possible to detect connective and bone tissue [13]. However, this result also raises the question of whether it would not be possible to specifically differentiate the dental follicle cells into osteoblasts and avoid differentiation into fibroblasts.

To understand differentiation, the stem cell niche model, or rather molecular processes inside and outside the cell, must be examined in detail [29,30,31,32,33,34]. Although a number of molecules involved in osteogenic differentiation have been discovered, they need to be understood in context. The differentiation of dental follicle cells under in vitro conditions is induced by external growth factors. Examples include bone morphogenetic factor 2 (BMP2) and insulin growth factor 2 (IGF2) [14,35,36]. In addition, differentiation depends on the composition of the extracellular matrix. Matrix proteins, such as collagen I and laminin, are of particular importance here. While collagen I induces early markers, such as alkaline phosphatase, in dental follicle cells, processes that favor mineralization will be aided by laminin [37,38].

### Discovery of the BMP2/DLX3 Signaling Pathway

In order to discover further mechanisms, proteomics and transcriptomics were carried out [39,40,41,42]. In these analyses, differentially expressed genes were identified before and after the induction of osteogenic differentiation. Bioinformatics studies revealed that very few genes or proteins were previously associated with osteogenic differentiation, including BMP2, IGF2, and proteins/genes associated with the WNT signaling pathway [43]. Other proteins that are associated with the BMP signaling pathway, such as SMAD proteins, were also found [40]. We gave our full attention to this signaling pathway, along with the transcription factor DLX3, which was previously known to be regulated by BMP2 and involved in bone marrow-derived osteogenic progenitor cell differentiation and tooth development [2,44,45,46].

As expected, initial investigations showed that the expression of DLX3 is induced by BMP2 [41]. Interestingly, upstream of DLX3, BMP2 not only activates SMAD proteins, but also protein kinase A (PKA), which, among other things, phosphorylates the β-catenin protein, which is a target protein of the WNT signaling pathway. After induction of osteogenic differentiation, β-catenin-like SMAD migrates into the cell nucleus. Here, β-catenin binds to its cofactor Lef, which in turn binds to the promoter of DLX3-like SMAD1/5 [47]. These transcription factors induce the expression of DLX3, which not only participates in the expression of the osteogenic transcription factor RUNX2 [46], but also induces the expression of BMP2, resulting in a positive feedback loop [41]. On the other hand, contrary to what is assumed after β-catenin induction, canonical WNT signaling appears to have an inhibitory effect on DLX3 expression and osteogenic differentiation of DFCs, although how WNT affects BMP2/DLX2 signaling is unknown [47]. In addition, BMP2 induces the activity of AKT (protein kinase B) [48]. However, the AKT signaling pathway mediates BMP2-dependent induction of the alkaline phosphatase activity (osteogenic differentiation) via induction of the transcription factor EGR1 (Early growth response protein 1) [48]. Interestingly, the parathyroid hormone-related protein (PTHrP) is secreted during osteogenic differentiation and has an inhibitory effect on early markers of osteogenic differentiation, such as ALP activity, but not on late markers, such as mineralization [49]. While these signaling proteins are associated with the DLX3/BMP2 signaling pathway and are more likely to affect differentiation at an early stage, factors that may affect osteogenic differentiation at a later stage are presented below. One of these factors is protein kinase C.

## 2. Protein Kinase C (PKC) Signaling and Biomineralization

### 2.1. PKC in Osteogenic Progenitor Cells

One factor appears to be PKC, which can be divided into several groups, but only classical PKCs are involved in the osteogenic differentiation of DFCs [50]. While the mechanism of the PKC signaling pathway, which is described below, is well known, little is known about its role in the osteogenic differentiation of DFCs. Interestingly, previous studies using both immortalized cell lines and somatic stem cells have demonstrated the involvement of PKC in osteogeogenic differentiation, with these studies showing that PKC can have both promoting and inhibitory effects. Miraoui et al. [51] showed that fibroblast growth factor receptor 2 (FGFR2)-induced osteoblast differentiation in murine mesenchymal C3H10T1/2 depended on PKC activation. In contrast, Nakura et al. have shown that PKC inhibits osteogenic differentiation in mouse preosteoblastic cell line MC3T3-E1, but promotes cell proliferation [52]. Similar conclusions were drawn with C2C12 cells by Lee and co-workers [53], who showed that PKC inhibits osteogenic differentiation by regulating the transcription factor MSX2, which in turn inhibits the expression of the osteogenic transcription factor RUNX2. The more recent studies with somatic stem cells also showed similar results. Li et al. [54], for example, showed that the inhibitory effect of miR-26a-5p on osteogenic differentiation of murine adipose-tissue-derived mesenchymal stem cells depends, among other things, on PKC inhibition. However, a study by Lotz et al. showed that PKC inhibition in human bone marrow-derived stem cells increased osteocalcin expression, but inhibited BMP2 expression [55]. Results from another study showed that activation of PKC, specifically, PKCβ1, resulted in repression of muscle ring finger protein-1 (MURF1)-mediated ubiquitylation of the peroxisome proliferator-activated receptor γ2 (PPARγ2) transcription factor. Stabilized PPARγ2 proteins enhanced adipogenesis and consequently reduced osteoblastogenesis from MSCs, showing that PKC activation suppresses osteogenic differentiation [56]. Similar inhibitory effects of PKC on osteogenic differentiation of PDL stem cells have recently been described. Wang et al. showed that Advanced glycation end product (AGE proteins) impaired the osteogenic potential via PKCβ2 [57].

### 2.2. PKC Signaling in DFCs

First experiments with DFCs showed that the expression of classical PKCs, e.g., PKCα, is inhibited from day 7 of osteogenic differentiation, but these experiments also showed that manipulating PKC activity had little effect on ALP activity, which is an early marker of osteogenic differentiation peaking at day 7 [50]. On the other hand, inhibition of classical PKCs supports the mineralization of DFCs, and this is the case even if PKCs were inhibited only a few days, but later than 1 week, after induction of differentiation, since inhibition of PKC occurring in the first week of osteogenic differentiation had no impact on the mineralization [50]. These results suggest that PKC does not affect the early phase of differentiation. Therefore, the interaction of PKC with AKT, which had a positive influence on osteogenic differentiation in previous work [48], was investigated [50]. Interestingly, and in contradiction to this previous work, in this study, the induction of differentiation with both BMP2 and dexamethasone appears to inhibit AKT expression/activity [50]. However, after inhibition of PKC, induction of osteogenic differentiation leads to activation of AKT [50]. Thus, AKT seems to play a complex role in differentiation, which is why regulation of AKT activity during differentiation has been investigated. These experiments showed that AKT does not seem to have a direct impact on mineralization, since both the inhibitor MK2206 and the activator SC-79 inhibited mineralization in a dexamethasone-based differentiation medium [50]. If one assumes that, as shown in this publication, PKC is essential for the activation of AKT, conversely, it should be investigated whether simultaneous inhibition of PKC and activation of AKT influences mineralization. When BMP2 was used for osteogenic differentiation, a stimulatory effect on the differentiation due to AKT inhibition was observed. These studies show that AKT is involved in the mechanism of differentiation. However, the importance of AKT on further signaling pathways, such as the BMP pathway, is ambiguous and depends on the cell line. Two recent studies on DFCs during osteogenic differentiation showed contradictory results regarding the activation of the BMP signaling pathway during differentiation. While Pieles et al. hypothesized an inhibitory effect of AKT on the BMP signaling pathway, Viale-Bouroncle and colleagues showed that AKT supports the BMP signaling pathway. These conflicting results suggest that AKT does not directly affect BMP signaling or that other, as yet unknown, signaling pathways influence AKT’s impact on BMP signaling. On the other hand, however, AKT activation seems to be essential for the inactivation of GSK-3β as a factor involved in the induction of β-catenin of the WNT pathway [50]. The canonical WNT signaling pathway and, in particular, its transcription factor β-catenin, which is also induced by BMP2 in DFCs, play a role in the early phase of DFC differentiation [47,58]. While PKCs have a direct impact on non-canonical WNT signaling activation, the active form of β-catenin is induced after PKC inhibition [50]. Since there have been results in recent years on the importance of the canonical WNT signaling pathway in DFCs, which had both a promoting and an inhibitory effect on osteogenic differentiation [47,58,59,60], this signaling pathway appears to have a modulating property for differentiation similar to the kinase AKT, which is characterized by the constitution of the individual DFC cell line. However, studies suggest that classical PKCs and canonical WNT signaling have a similar inhibitory effect on osteogenic differentiation of DFCs [47,50]. It will, therefore, be important for understanding differentiation to uncover by which biological processes classical PKCs are linked to the canonical WNT signaling pathway in DFCs during osteogenic differentiation.

A signaling pathway that is of importance for cell differentiation is the NFκ-B signaling pathway. This signaling pathway both inhibits and promotes the differentiation of osteogenic progenitor cells and is inhibited in dental follicle cells after induction of differentiation [61,62,63,64]. However, it appears to have an inhibitory effect on osteogenic differentiation of DFCs, as decreased mineralization was shown after NFκ-B induction with NFκ-B activator PMA. Interestingly, PMA activates NFκ-B, but independently of classical PKC [65]. Novel PKCs, which do not influence osteogenic differentiation of DFCs, can directly activate the NFκ-B signaling pathway [65]. However, a previous study showed that classical PKCs caused the activation of RELB of noncanonical NF-κB signaling, but not RELA of canonical NF-κB signaling, in cancer cells [66]. So, this could be an explanation why it was not possible to induce osteogenic differentiation markers by inhibiting the canonical NFκ-B signaling pathway. Interestingly, regulation of AKT has only little influence on the expression of NFκ-B signaling pathway proteins after induction of osteogenic differentiation with dexamethasone. On the other hand, NF-κB can be induced by AKT, and both signaling pathways play important roles in cell viability [67]. We believe that the NFκ-B signaling pathway does not play a significant role in osteogenic differentiation. Previous studies on the proteome and transcriptome did not find any significant evidence that the NFκ-B signaling pathway is involved in differentiation [68]. It is possible that PMA induces PKC, but impairs cell viability or cell proliferation, as well [69], and it is therefore possible that PMA suppressed the mineralization of DFCs by impairing cell viability without activation of NFκ-B signaling, which could support cell viability [70]. This conclusion is supported by the following observation from a previous study. The inducer of the non-canonical WNT signaling pathway WNT5A, which supports cell viability during osteogenic differentiation [71], inhibited the activation and expression of classical PKCs in DFCs. The results presented here can be summarized as follows [50]. PKCs inhibit osteogenesis specifically by regulating the kinase AKT, thereby affecting the downstream activity of β-catenin and the NF-κB signaling pathway (Figure 1).

The question, therefore, arises to what extent do PKCs influence crucial biological processes or are influenced by them, and what significance do these processes have for the osteogenic differentiation? If we look at the mechanism for the activation of classical PKC, it is striking that this cascade starts with the activation of a G protein-coupled receptor (GPCR), which activates phospholipase C (PLC) downstream. Activation of PLC cleaves the lipid phosphatidylinositol-4,5-bisphosphate (PIP2) by hydrolysis into diacylglycerol (DAG) and inositol-1,4,5-trisphosphate (IP3), which releases calcium as another cofactor for the activation of PKC [72,73]. DAG also plays an important role in the activation of PKC as a cofactor [73], and its presence in the lipidome is closely related to the synthesis of lipids [74]. While phosphorylation also plays an important role in PKC activation, the interaction of classical PKC and DAG demonstrates the connection between signaling pathways and crucial biological processes, such as lipid metabolism, which will be discussed below in its closer relation to osteogenic differentiation of DFCs.

## 3. Fatty Acid Synthesis and the Lipidome during Osteogenic Differentiation

Almost no attention has been brought so far to the fatty acid metabolism and the composition of the lipids whose synthesis is related to fatty acid metabolism during osteogenic differentiation of DFCs or other types of dental stem cells. This is not only of particular interest, as diacylglycerides (DAG) are a cofactor of classical PKC, but also because previous studies have shown that lipid composition changes in stem cells [75,76,77,78].

### 3.1. Fatty Acid Synthesis

Fatty acid synthesis is an important part of metabolism and crucial for developmental processes [34,79,80,81,82]. Saturated fatty acids, like palmitic acid, can be distinguished from unsaturated fatty acids, like linolenic acid. Previous studies showed that dramatic changes in the composition of fatty acids and lipids take place in stem cells, suggesting their importance for almost all biological processes, including differentiation and cellular senescence [34,75,77,78,83]. Levental et al. showed that cell culture supplementation with docosahexaenoic acid (DHA), a lipid component characteristic of osteoblast membranes, induced extensive lipidome remodeling in MSCs [84]. These changes induced by DHA supplementation enhanced osteogenic differentiation of bone marrow stem cells, with simultaneous enhanced AKT activation. Levental et al. proposed a model that membrane microdomains increase AKT activity and thereby enhance osteogenic differentiation [84]. It is generally known that fatty acid synthesis is linked to energy metabolism and, in particular, to the citric acid cycle [34]. Citric acid can also be seen as the starting molecule for fatty acid synthesis, which is broken down into acetic acid by means of ATP citrate lyase. The acetic acid is converted to acetyl-CoA via acetyl-CoA synthetase, from which malonyl-CoA is then formed with the aid of acetyl-CoA carboxylase. Acetyl-CoA carboxylase is a key enzyme that can regulate fatty acid synthesis after phosphorylation. The malonyl-CoA is the starting molecule of the fatty acid synthetase, which forms the long-chain fatty acids. We studied the expression of these enzymes during differentiation to predict the formation of fatty acids during differentiation [74]. The expression of the acyl-CoA synthase, which is involved in the synthesis of lipids, among other things, was also examined. If we now look at protein expression, we see an induction of the most important proteins of fatty acid synthesis, especially in the late phase of differentiation [74]. The phosphorylated and inhibitory form of acetyl-CoA carboxylase is significantly down-regulated. On the other hand, the acyl-CoA synthetase is only very weakly regulated, which suggests a balanced metabolism of lipids. Investigations into the influence of fatty acid synthesis on osteogenic differentiation demonstrated that this biological process plays an important role, which prompted us to study the composition of lipids during differentiation [74].

### 3.2. Lipidome

Now, we turn our attention to the changes in lipids during differentiation. In the last few years, a number of papers on the subject of lipidomics and differentiation have been published [34,75,83,85,86,87,88,89]. Even if we cannot expect any functional insight into the role of lipids in differentiation with this analysis, as with all omics analyses, we can make initial vague assumptions about it [68]. Lipid molecules consist of a polar group, which is the hydrophilic part of the molecule, and by which the diacylglycerides can be divided into different groups. Mention should be made here of ethanolamine, inositol, choline, and serine. Then there are the fatty acids already mentioned and, as a third part, the triol, glycerol. For example, if you combine choline or serine with glycerophosphoric acid and two fatty acids, you obtain phosphatidylcholine or phosphatidylserine, respectively, which are both important components of the cell membrane and cell signaling. In the lipidome analysis, it was noticeable that the proportion of lipids compared to the protein concentration had increased significantly during differentiation [74]. A closer look at the composition of the lipids first shows that the proportion of phosphatidylserines has hardly changed. Phosphatidylserines are part of the cell membrane and play some role in apoptosis, which is tightly regulated during differentiation in DFCs [41,71]. This could be a reason for the constant proportion of these lipids. Things are quite different with phosphatidylethanolamines, which are also important components of the cell membrane and induced during osteogenic differentiation. Due to its influence on the fluidity of the cell membrane, it is also important for the morphology of the cell. However, on the basis of further data, statements can only be made about a greatly changed composition of the phosphatidylethanolamines, which in turn speaks for an increased synthesis of these molecules. Phosphatidylethanolamine has been reported as a key participant in several biological processes, such as autophagy, which inhibits the osteogenic differentiation of DFCs [90], and the modulation of cell membrane properties [91]. Phosphatidylethanolamines were also significantly increased in osteoblasts compared to undifferentiated stem cells [84], but the significance of phosphatidylethanolamine changes for differentiation is not known. Important lipids of the cell membrane are also the phosphatidylcholines and the phosphatidylinositiols. The concentration of phosphatidylcholines remained almost unchanged during osteogenic differentiation in DFCs [74], which was similar to investigations with adipose tissue-derived mesenchymal stem cells after osteogenic differentiation [78]. A significant increase in the ratio of phosphatidylinositiols to phosphatidylserines during differentiation could be determined (Figure 2), which could also possibly be related to the inhibition of PKC expression since phosphatidylserine concentrations correlate well with PKC activity [92]. PI is associated via PI3K with the AKT signaling pathway, whose importance for the differentiation of DFCs was described above. Moreover, Kilpinen et al. showed that the ratio of phosphatidylinositol to phosphatidylserine increased toward the end of extensive expansion of human stem cells from the bone marrow, which correlated with reduced functionality as stem cells [77]. Interestingly, however, phosphatidylserine promotes osteogenic differentiation and is particularly involved in mineralization in human bone marrow-derived mesenchymal stem cells [93], which could explain the relatively low concentration of phosphatidylserine during osteogenic differentiation in the cell membrane of DFCs. It can be speculated (Figure 3) that phosphatidylserine is a component of small extracellular vesicles, which contain not only calcium and phosphates, but also enzymes, such as alkaline phosphatase, and could be responsible for the initiation of mineralization [94,95]. Cruz et al. showed that phosphatidylserine nucleated amorphous calcium phosphate that converted into biomimetic apatite [95]. Other groups found during lipidome analyses of DFCs are sphingomyelins and cholesterols, both of which are involved in a number of processes and cell membrane stability [74]. The metabolism of sphingomyelins is involved in bone development by induction of signaling pathways, such as the BMP signaling pathway in osteogenic progenitor cells [96,97], which is, as explained above, also induced during the osteogenic differentiation of DFCs. Interestingly, cholesterols play a different role during osteogenic differentiation. If we compare the low level of esterified cholesterol during differentiation, it seems that a constant level of free cholesterol is important for differentiation. Since free cholesterol increases cell membrane fluidity, morphological flexibility is probably important for the differentiation process. However, the total concentration of cholesterol per cells decreases during osteogenic differentiation of DFCs [78], because cholesterol inhibits osteogenic differentiation [98], and impaired cholesterol synthesis improves osteogenic differentiation [99]. Another study showed that inhibition of cholesterol synthesis induces osteogenic differentiation via regulation of Rho-GTPases, which increases cell rigidity [100]. However, a lower cholesterol concentration also correlates well with a lower viscosity of the cells. This part of the lipidome appears to have a strong impact on decreased cellular stiffness [78] that can positively affect the osteogenic differentiation of DFCs [101]. Other parts of the lipidome that have changed significantly during differentiation are the diglycerides and triglycerides, which have increased relative to the control during differentiation in differentiating cells [74]. Since triglycerides are known to store energy in the cell, this is probably related to increased energy production during differentiation. One can also assume an increased proportion of adipogenic cells during in vitro differentiation, since dexamethasone is also used to induce adipogenic differentiation. In addition, previous studies showed that the inductive effect of melatonin on the osteogenic differentiation is based on an inhibition of the accumulation of triglycerides and the induction of oxidative phosphorylation, which is associated with β oxidation of fatty acids [102,103]. A strongly controlled synthesis of new lipids can therefore be assumed, which also suggests a constant expression of the acyl-CoA synthetase during the differentiation of DFCs [74].

## 4. Energetic Metabolism during Osteogenic Differentiation

That energy metabolism plays a role in osteogenic differentiation seems to be true due to the fact that autophagy and AMP kinase, which are closely related to the regulation of energy metabolism, can impede differentiation [90]. Transcriptome and proteome studies revealed overrepresented signaling pathways and biological processes associated with carbohydrate metabolism and, in particular, with glycolysis after induction of differentiation [68]. However, previous studies with other osteogenic progenitor cells have shown that these cells prefer oxidative phosphorylation for energy supply during differentiation [104,105,106]. Interestingly, Smith and Eliseev [106] recently showed that induction of the oxygen consumption rate is associated with activation of BMP2 and AKT, which are also involved in differentiation in DFCs [9]. Moreover, Kumar et al. showed that defects in energy metabolism are associated with functional exhaustion of bone marrow mesenchymal stem cells [107]. It, therefore, seems more than obvious to discuss the energy metabolism during osteogenic differentiation of DFCs [74].

The most important processes of a mammalian cell, which serve to generate energy, can be divided into three phases: glycolysis, citric acid cycle, and oxidative phosphorylation. The very simplified Figure 4 shows how, in the cytoplasm, a glucose molecule is metabolized into a molecule of pyruvate, which in the mitochondrion further becomes acetyl-CoA, which then enters the citric acid cycle. Not only are NADH molecules obtained here, for example, which can be used in oxidative phosphorylation to obtain energy through ATP synthesis, but their intermediate products, such as citric acid, can be exported back into the cytoplasm and used as starting materials for the synthesis of glucose (gluconeogenesis) and fatty acids. Fatty acid or esterified fatty acids then serve as energy stores and can be metabolized by the cell for energy production when required, which has already been mentioned above about the complex lipidome of the DFCs in osteogenic differentiation [74].

Glycolysis simply converts a six-carbon atom glucose molecule into two pyruvate molecules, which only consist of three carbon atoms. During this process, energy is gained through the formation of ATP. Through the production of pyruvate or after a reaction of the pyruvate dehydrogenase with the acetyl-CoA, further processes for energy production (citric acid cycle and oxidative phosphorylation) can be started under normoxic conditions. Under hypoxic conditions, however, the pyruvate is not used by the subsequent steps for further energy production. In contrast, it is metabolized by lactate dehydrogenase into lactic acid, which can then be exported from the cells. However, this reaction also takes place under normoxic conditions, which is why the extracellular lactate concentration is often a good indicator of glycolysis. After induction of differentiation, the lactate concentration was increased. An important step in glycolysis is hexokinase, whose activity was also increased after 7 days of induction of osteogenic differentiation, which is probably due to increased protein expression [74]. It is striking here that this enzyme was induced precisely in the late phase of differentiation [74]. There is also increased expression of phosphofructokinase, which is the next step in glycolysis. On the other hand, protein expressions of the downstream enzymes GAPDH and pyruvate kinase were not significantly increased after differentiation [74]. However, the different expressions of the two enzymes that further metabolize the pyruvate are interesting. While lactate dehydrogenase, which is induced particularly under conditions when little energy can be obtained via oxidative phosphorylation, was hardly induced after differentiation, pyruvate dehydrogenase was strongly induced in the differentiation media (particularly with dexamethasone) compared to the DMEM control [74]. These results suggest that glycolysis is an important step in energizing DFCs during differentiation. Glycolysis also appears to be an essential process for osteogenic differentiation in general. Studies with different types of stem cells have shown that induction of glycolysis during osteogenic differentiation leads to increased expression of osteogenic markers [108,109]. Aerobic glycolysis (Warburg effect) is not the main energy source in the osteogenic differentiation [105,110], unlike recently shown in induced pluripotent stem cell-derived mesenchymal stem cells (iPSC-MSCs) [109]. Moreover, too high a glucose concentration, as occurs in cells of diabetics, can lead to inhibition of differentiation through the formation of reactive oxygen species (ROS), which, interestingly, depends on the formation of Nicotinamide adenine dinucleotide phosphate (NADPH) [111].

This conclusion indicates that energy is mainly gained through oxidative phosphorylation during differentiation. If this is the case, this should be reflected in the expression of the enzymes of the citrate cycle. In DFCs, during osteogenic differentiation, the expression of aconitase, isocitrate dehydrogenase, and all other enzymes in this cycle was induced [74]. An induction of the citrate cycle is also confirmed by the induction of malate dehydrogenase activity [74]. After the induction of the citric acid cycle indicated an increased concentration of NADH that is required for the generation of energy through oxidative phosphorylation, subsequent investigations showed that the number of mitochondria in the cells also increased during differentiation [74]. For example, the protein expression of typical mitochondrial proteins, such as cytochrome C or the mitochondrial pyruvate carrier proteins, increased in late phases of differentiation. The increase in mitochondria could also be shown by a significant increase in mitochondrial DNA in the cells. However, it was also interesting that in an early phase of differentiation, before the increase in mitochondria in the cells could be detected, energy production through oxidative phosphorylation increased, which was detectable through the induction of complex I activity. This early induction of oxidative phosphorylation is also indicated by a significant reduction in the expression of the hypoxy marker Hif1α [74]. This observation is in contrast to the study by Mao et al. with PDL cells. This work demonstrated that induction of a hypoxy-like state in PDL cells induces osteogenic differentiation [108]. However, in conclusion, the results on glycolysis, the citric acid cycle, and oxidative phosphorylation indicate an increased energy requirement in DFCs, which the cells cover through increased oxidative phosphorylation. Overall, oxidative phosphorylation seems to play a more important role in osteogenic differentiation than in other cell differentiations, e.g., in chondrogenic differentiation [112]. Recent studies show that the mitochondria are of essential importance for osteogenic differentiation [103,110,113]. It seems that not only the number of mitochondria is important [110], but also the morphology of the cell organelle [113], which is associated with the activity.

A byproduct of these metabolic changes has been shown to be an increase in oxidative stress. The formation of high concentrations of ROS is very harmful to the process of osteogenic differentiation, but it can be mitigated by induction of enzymes involved in the degradation of ROS [112]. These enzymes include catalase, which is induced during osteogenic differentiation of DFCs. Interestingly, other stress proteins, such as SOD1 or thioredoxin, are poorly regulated during differentiation. However, an experiment with a catalase inhibitor (3-AT) showed that the control of ROS is essential for differentiation. In this context, it is also interesting to point out the links between PKC and AKT with the induction of ROS. It could be shown in DFCs that an inhibition of AKT leads to a reduced formation of ROS. It is also important to note that not only can PKC be activated by ROS [114], but also that PKC itself activates processes leading to the formation of ROS [115]. In this context, the inhibition of PKC during differentiation must be understood. During osteogenic differentiation, there seems to be an association between metabolic processes and PKC or AKT, which has not yet been sufficiently investigated in DFCs.

## 5. Conclusions

Our review article summarized recent developments in research on osteogenic differentiation in DFCs. In this article, we raised new questions about the relationship between classical PKCs and AKT and other signaling pathways, such as the BMP pathway and the canonical and non-canonical WNT pathways. These relationships seem complex and, e.g., the influence of PKC and AKT on the BMP signaling pathway is also dependent on the disposition of the individual cell line before induction of osteogenic differentiation. However, a causal relation between inhibition of PKC and activation of AKT is almost certain after induction of osteogenic differentiation with BMP2. AKT appears to support at least the early phase of osteogenic differentiation. AKT is also activated at later stages of osteogenic differentiation and, here, likely supports the induction of oxidative phosphorylation with an increased rate of oxidative stress, although induction of AKT did not improve the mineralization of DFCs. Data from the lipidome and proteome support the observed regulation of PKC and AKT during osteogenic differentiation. Changes of the lipidome during osteogenic differentiation also suggest high energy consumption, oxidative stress defense, and fundamental changes in the viscosity and fluidity of the lipid membrane. Although we have revealed here that there is an intimate relationship among PKC, AKT, and the metabolic processes in DFCs, other signaling pathways and biological processes are likely involved that are still unknown. It will also be interesting to what extent extracellular vesicles influence this process [25,116,117]. Identifying and controlling these unknown processes will be an important step in applying the mechanisms of differentiation for therapeutic purposes.

## Figures and Tables

**Figure 1 biomedicines-11-02787-f001:**
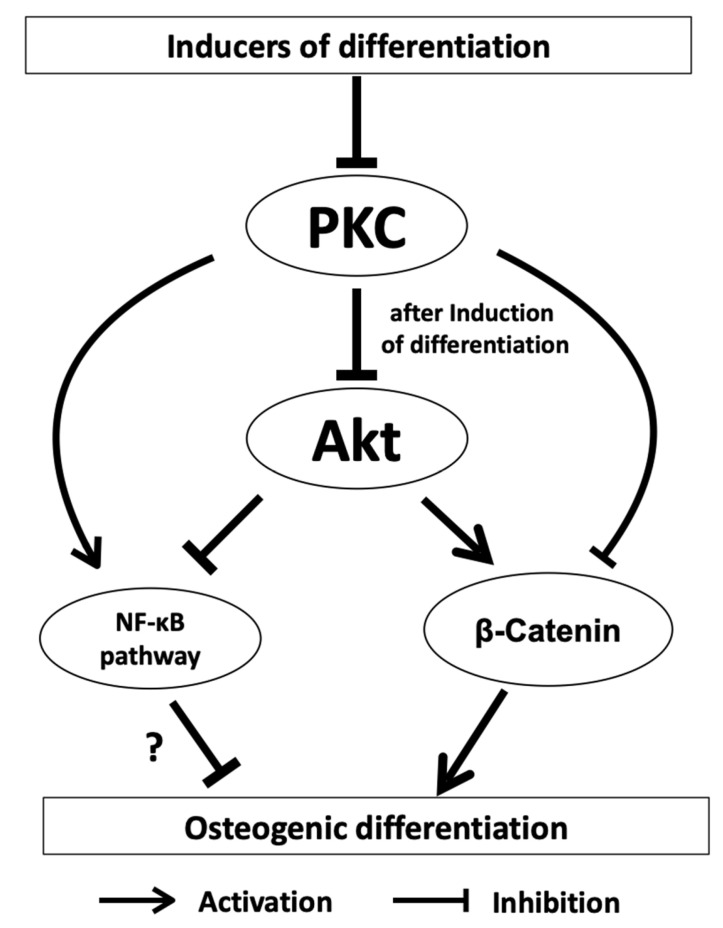
PKCs inhibit osteogenesis by regulating the kinase AKT, which influences downstream β-catenin and the NF-κB signaling pathway. The effect of the NF-κB signaling pathway on osteogenic differentiation is probably small (?).

**Figure 2 biomedicines-11-02787-f002:**
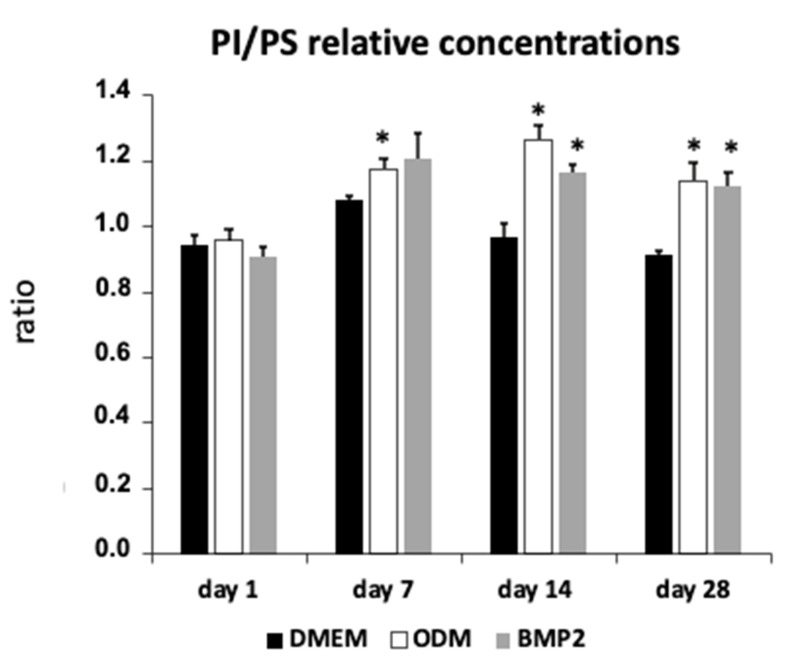
The diagram shows the relative concentrations of phosphatidylinositiols (PIs) to phosphatidylserines (PSs) identified during osteogenic differentiation in DFCs. Cells were cultured for the indicated periods in osteogenic differentiation medium, containing either bone morphogenetic protein 2 (BMP2) or dexamethasone (ODM) as an inducer, or in a control medium (DMEM) without an inducer. The data come from a study that analyzed the lipidome on the days shown [74]. PI/PS Ratio is inversely correlated with the activity of PKC. Please refer to Figure 3 for further explanation. Bar charts represent the mean +/− standard deviation (*n* = 3). The stars mark significant differences from the control group DMEM (Student’s test); *: *p* < 0.05.

**Figure 3 biomedicines-11-02787-f003:**
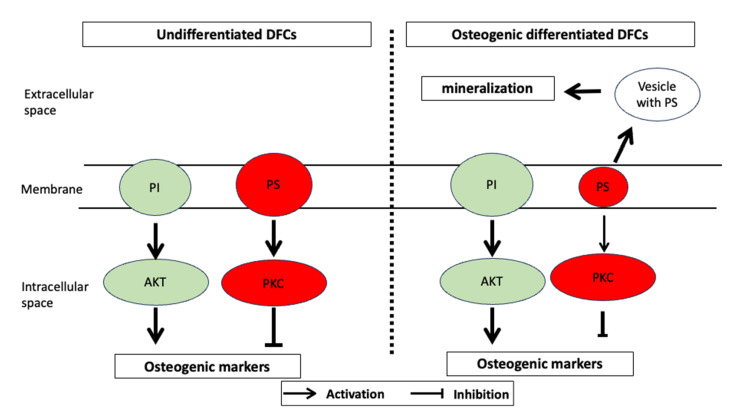
Hypothetical model showing the relative concentrations of PI and PS in the cell membrane and the postulated effects on AKT and PKC in undifferentiated (**left**) and differentiated (**right**) DFCs. The increasing PI/PS ratio after differentiation induction (Figure 2) activates AKT and inhibits PKC, which has a stimulatory effect on differentiation. In this model, it is also assumed that the lower PS concentration is due to increasing extracellular PS concentration in mineralizing vesicles, leading to increased mineralization.

**Figure 4 biomedicines-11-02787-f004:**
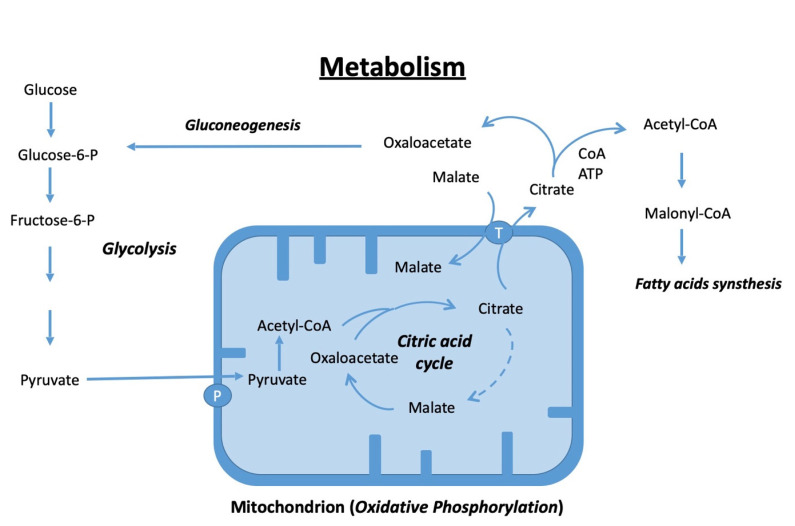
Simplified sketch of the relationships of glycolysis, the citrate cycle, oxidative phosphorylation (production of energy and oxidative stress), gluconeogenesis, and fatty acid metabolism in a cell. P: mitochondrial pyruvate carrier (transport of pyruvate into mitochondrion); T: tricarboxylate transport protein (malate transport into and citrate out of the mitochondrion).

## Data Availability

Not applicable.

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
