# Peer review of "Evaluation of Current Studies to Elucidate Processes in Dental Follicle Cells Driving Osteogenic Differentiation"

_biomedicines, 2023, doi:10.3390/biomedicines11102787_

Round 1

Reviewer 1 Report

The review “Evaluation of current studies to elucidate processes in dental follicle cells driving osteogenic differentiation” by Morsczeck et al is an interesting analysis of osteogenic differentiation studies of dental follicle cells. The authors analyze current literature and make several original conclusions. The review is well and clear written, but the manuscript would benefit from some minor changes.

1.     The paragraphs of long text would better be divided by subheadings, thus making the text more understandable

2.     When describing PKC signaling the Notch signaling and its relation to PKC in osteogenic differentiation should be mentioned

3.     The manuscript would benefit if the authors add some schemas of the interactions between the signalings as they discuss it in the section 2 and 3

4.     The legends to the figures 1 and 2 should be more comprehensive and explain the figures in details

5. The comparison to other osteogenic cells would be usuful in such a review

The English is OK

Author Response

  1. The paragraphs of long text would better be divided by subheadings, thus making the text more understandable

Answer to point 1: We have put some subheadings. However, this was not possible for all chapters of this work.

  1. When describing PKC signaling the Notch signaling and its relation to PKC in osteogenic differentiation should be mentioned

Answer to point 2: We cannot find any article about this relation and we have not tackled this question yet

  1. The manuscript would benefit if the authors add some schemas of the interactions between the signalings as they discuss itin the section 2 and 3

Answer to point 3: Two new figures are included that will support readers

  1. The legends to the figures 1 and 2 should be more comprehensive and explain the figures in details

Answer to point 4: Figure legends were revised according to reviewer’s suggestion

  1. The comparison to other osteogenic cells would be usuful in such a review

Answer to point 5: Thank you for this suggestion. We sought to compare the mechanisms described here for DFCs, such as the role of PKC during differentiation, with other cells. However, implementing this for all the processes presented in this article is a completely new challenge. A whole new article needs to be written for this. It's already on my to-do list

Reviewer 2 Report

The review article by Morsczeck et al. sumarizes current knowledge in the field of osteogenic differentiation of mesenchymal stem cells with the focus on dental foliculle cells (DFCs). The special emphasis is put on the processes such as signalling pathways and metabolic processes during the osteogenic differentiation of DFCs. The findings and conclusions in the review are also supported by several previous articles published by authors.
The text is well written and comprehensible for readers.

I have several minor comments/questions for the authors before the manuscript is accepted:
- Abstract - "Akt" and "AKT". This should be unified.

- page 2 - row 76 - "(BMP)2" vs. "(BMP2)" and "(IGF)2" vs. "(IGF2)"

- Fig. 1 - It seems that the expansion of abbreviations (DMEM, ODM, BMP2) used in the figure is missing (though it can be found in the cited publication). It would be more comprehensible for readers to include the spelled-out form of these in the figure caption as well.

- To what extent can the described cell processes during osteogenic differentiation be generalized to other (mesenchymal) stem cell types?

The text is well written and comprehensible for readers.

Author Response

  1. I have several minor comments/questions for the authors before the manuscript is accepted:
    - Abstract - "Akt" and "AKT". This should be unified.
    Answer to point 1: Thank you for this comment. We changed “Akt” into “AKT”.

  2. page 2 - row 76 - "(BMP)2" vs. "(BMP2)" and "(IGF)2" vs. "(IGF2)"

Anwer to point 2: Thank you for this advice. We revised manuscript

3. Fig. 1 - It seems that the expansion of abbreviations (DMEM, ODM, BMP2) used in the figure is missing (though it can be found in the cited publication). It would be more comprehensible for readers to include the spelled-out form of these in the figure caption as well.

Answer to point 3: Figure legend was revised according to reviewer’s suggestion

  1. To what extent can the described cell processes during osteogenic differentiation be generalized to other (mesenchymal) stem cell types?

Answer to point 4: Thank you for this suggestion. We sought to compare the mechanisms described here for DFCs, such as the role of PKC during differentiation, with other cells. However, implementing this for all the processes presented in this article is a completely new challenge. A whole new article needs to be written for this. It's already on my to-do list